# Design of a Microbial Remediation Inoculation Program for Petroleum Hydrocarbon Contaminated Sites Based on Degradation Pathways

**DOI:** 10.3390/ijerph18168794

**Published:** 2021-08-20

**Authors:** Xingchun Li, Wei He, Meijin Du, Jin Zheng, Xianyuan Du, Yu Li

**Affiliations:** 1State Key Laboratory of Petroleum Pollution Control, Beijing 102206, China; lixingchun@163.com (X.L.); zhengjin2810@163.com (J.Z.); duxianyuan@cnpc.com.cn (X.D.); 2CNPC Research Institute of Safety and Environmental Technology, Beijing102206, China; 3MOE Key Laboratory of Resources Environmental Systems Optimization, North China Electric Power University, Beijing 102206, China; v493287@163.com (W.H.); mjdu0401@outlook.com (M.D.)

**Keywords:** petroleum hydrocarbons, remediation of petroleum contaminated sites, microbial remediation, molecular dynamics modelling, Copeland method

## Abstract

This paper analyzed the degradation pathways of petroleum hydrocarbon degradation bacteria, screened the main degradation pathways, and found the petroleum hydrocarbon degradation enzymes corresponding to each step of the degradation pathway. Through the Copeland method, the best inoculation program of petroleum hydrocarbon degradation bacteria in a polluted site was selected as follows: single oxygenation path was dominated by *Streptomyces avermitilis*, hydroxylation path was dominated by *Methylosinus trichosporium* OB3b, secondary oxygenation path was dominated by *Pseudomonas aeruginosa*, secondary hydroxylation path was dominated by *Methylococcus capsulatus*, double oxygenation path was dominated by *Acinetobacter baylyi* ADP1, hydrolysis path was dominated by *Rhodococcus erythropolis*, and CoA path was dominated by *Geobacter metallireducens* GS-15 to repair petroleum hydrocarbon contaminated sites. The Copeland method score for this solution is 22, which is the highest among the 375 solutions designed in this paper, indicating that it has the best degradation effect. Meanwhile, we verified its effect by the Cdocker method, and the Cdocker energy of this solution is −285.811 kcal/mol, which has the highest absolute value. Among the inoculation programs of the top 13 petroleum hydrocarbon degradation bacteria, the effect of the best inoculation program of petroleum hydrocarbon degradation bacteria was 18% higher than that of the 13th group, verifying that this solution has the best overall degradation effect. The inoculation program of petroleum hydrocarbon degradation bacteria designed in this paper considered the main pathways of petroleum hydrocarbon pollutant degradation, especially highlighting the degradability of petroleum hydrocarbon intermediate degradation products, and enriching the theoretical program of microbial remediation of petroleum hydrocarbon contaminated sites.

## 1. Introduction

The rapid development of the petroleum industry and its products, and the increase in the use of petroleum fossil fuels have led to serious environmental pollution. The pollution can be largely attributed to petroleum hydrocarbon pollutants [1]. Petroleum hydrocarbon pollutants are classified as priority environmental pollutants [2], which primarily exist in the form of *n*-alkanes, branched-chain alkanes, cycloalkanes, and aromatic hydrocarbons in the environment [3,4]. Most of them can exist stably in the environment for a long time and are not easily degraded. Crude oil leakage is one of the fundamental causes of petroleum hydrocarbon pollution, which has caused great harm to the soil ecosystem [5,6,7]. The petroleum hydrocarbon pollutants can affect the physical and chemical properties (such as soil texture, compactness, osmotic resistance, and the concentration and content of heavy metals) of soil [8]. Petroleum hydrocarbon pollutant-enriched soil adversely affects the growth of plants and animals, and in some cases can even cause death, resulting in extensive ecosystem damage [9,10,11,12]. At the same time, petroleum hydrocarbon pollutants exhibit carcinogenic, neurotoxic, genus toxic, and other toxicological properties, which pose a serious threat to human health [13,14]. Therefore, the repair of petroleum-contaminated sites has become a focus of attention.

Petroleum hydrocarbon pollution has always been a concern in various countries, and standards have been set in various countries for petroleum hydrocarbon pollution. The US government has established regulations or guidelines for some petroleum hydrocarbon fractions and their components to protect the public from the potentially harmful health effects of petroleum compounds. In turn, all states have their own standards and methods for cleaning up petroleum hydrocarbons or their components [15]. For example, the Washington State Department of Ecology has set limits for soils of 120–6000 mg/kg (gasoline series organics), 240–6000 mg/kg (diesel series organics) and 3600–4400 mg/kg (residual petroleum hydrocarbons) [16]. Compared to developed countries such as the USA, UK, and Canada, Asia’s petroleum hydrocarbon contaminated land management systems lag behind in their effectiveness in identifying petroleum hydrocarbon contaminated sites, conducting appropriate detailed risk assessments and initiating remediation activities. Asian countries generally lack specific regulations for the management of petroleum hydrocarbon contamination of soil and groundwater. In those countries or regions that do not have any cleanup standards, or legal and regulatory frameworks in place to assess and manage petroleum hydrocarbon, contaminated sites still use the US EPA standards [17]. The latest Soil Environmental Quality Soil Contamination Risk Control Standard for Construction Land (Trial) (GB 36600-2018), promulgated by China in 2018, sets out risk screening values for soil contamination based on human health, but this standard only sets limit values for extractable petroleum hydrocarbons (C_10_ to C_40_). Under China’s current environmental management framework, there are still many gaps in the control of petroleum-based pollutants in soil, and the few contaminated sites that have been evaluated and remediated have had to draw on standards already in place in other countries. Therefore, further improving the delineation and establishing supporting analytical methods for petroleum hydrocarbon pollutants is one of the priorities for future environmental management in China.

To achieve effective remediation of petroleum contaminated sites, suitable remediation methods need to be selected based on the nature of the land type and extent of contamination of the petroleum contaminated sites [18,19,20]. Chemical drenching, electrokinetic remediation, solvent extraction, microwave radiation, and bioremediation are currently the commonly used methods to realize the remediation of petroleum-contaminated sites [21,22,23]. Among them, bioremediation is the most important method followed for the remediation of oil-contaminated sites. This method does not cause secondary pollution and is economical and effective in realizing remediation. Therefore, it has been used as a common remediation method to repair petroleum-contaminated sites [24]. The microbial remediation method, another of the commonly used bioremediation methods, can be used to remove petroleum hydrocarbon pollutants present. In this method, the microbial population in the environment is biodegraded [25]. This method is more economical and environmentally friendly than other remediation methods [26].

Petroleum hydrocarbon degradation bacteria first convert petroleum hydrocarbon pollutants to intermediates through the process of intracellular oxidation. In the next step, the products obtained in the previous step are used for the degradation reaction. Petroleum hydrocarbon pollutants are finally degraded into environmentally friendly water and carbon dioxide [27]. When the number of native petroleum hydrocarbon degradation bacteria in petroleum contaminated sites is low, exogenous microorganisms are inoculated into the contaminated sites to be remediated [28,29]. Mishra et al. [30] used *Pseudomonas aeruginosa* PSA5 and *Rhodococcus* sp. NJ2 for the studies and found that these bacteria could degrade *n*-hexadecane in petroleum sludge via the terminal oxidation pathway. The degradation rates reached 99% and 95% in 10 days. Zhang et al. [31] found that *Pseudomonas aeruginosa* DQ8 degraded 34.5% of pyrene via the singlet and dioxygen pathways in 12 days. Yu et al. [32] isolated and domesticated two highly efficient oil-degrading strains, Bacillus subtilis SWH-1 and Sphingomonas polyphila SWH-2, from different oil fields, and prepared them into highly efficient microbial formulations to be put into the contaminated site of Shengli oil field. However, most researchers have only investigated the remediation of petroleum hydrocarbon contaminated sites in the presence of one or more petroleum degradation bacteria following specific reaction steps in the degradation pathway. Results on the remediation of petroleum hydrocarbon contaminated sites in the presence of multiple petroleum hydrocarbon degradation bacteria, a complete analysis of the degradation pathway of the petroleum hydrocarbon degradation bacteria, and all the reaction steps have not been reported.

This paper presents the results of the analysis of the degradation pathways of petroleum hydrocarbon pollutants in the presence of petroleum degradation bacteria. A variety of degradation enzymes according to the degradation path was selected to design the inoculation program of the petroleum hydrocarbon degradation bacteria. Firstly, the molecular dynamics method was used to calculate the binding energy of the degradation enzymes and petroleum hydrocarbon pollutants in each step to determine the degradation ability. Following this, a Java program was developed to arrange and combine the degradation enzymes to develop a variety of petroleum hydrocarbon degradation bacteria inoculation programs. Finally, a variety of inoculation programs of petroleum hydrocarbon degradation bacteria were scored by the Copeland method, and the best inoculation program was selected according to the score of each program. Our aim is to design a petroleum hydrocarbon degradation bacterial inoculation program for the remediation of petroleum hydrocarbon contaminated sites. This study will consider the main pathways of petroleum hydrocarbon contaminant degradation and provide a new direction for further research on the microbial remediation of petroleum hydrocarbon contaminated sites.

## 2. Materials and Methods

### 2.1. Selection of Degradation Enzymes of Petroleum Hydrocarbon Degradation Bacteria

The degradation pathways of the three petroleum hydrocarbon pollutants (*n*-tetradecane, norphytane, and cyclopentane) were studied. It was found that the degradation pathways of *n*-tetradecane, norphytane, and cyclopentane similarly involved the participation of similar enzymes. The degradable proteins of the same petroleum hydrocarbon degradation bacteria were selected to study the degradation ability and the degradation pathways of *n*-tetradecane, norphytane, and cyclopentane. For the single oxygenation pathway, five types of monooxygenases, membrane-integrated methane monooxygenase of *Methylomicrobium alcaliphilum* 20Z (PDB ID: 6CXH) [33], flavin-dependent monooxygenase of *Cellvibrio* sp. BR (PDB ID: 4USQ) [34], flavin-dependent monooxygenase of *Nitrincola lacisaponensis* (PDB ID: 6HNS) [35], bacterial cytochrome P450 enzyme of *Bacillus megaterium* (PDB ID: 1SMJ) [36], and cytochrome P450 enzyme of *Streptomyces avermitilis* (PDB ID: 3E5K) [37] were selected. The hydroxylation pathway was conducted using five kinds of hydroxylases: methane monooxygenase hydroxylase of *Methylococcus capsulatus* (PDB ID: 1FZI) [38], methane monooxygenase hydroxylase of *Methylosinus trichosporium* (PDB ID: 1MHZ) [39], long-chain alkane monooxygenase of *Geobacillus thermodenitrificans* (PDB ID: 3B9N) [40], methane monooxygenase hydroxylase of *Methylosinus sporium* (PDB ID: 6D7K) [41], and methane monooxygenase hydroxylase of *Methylosinus trichosporium* OB3b (PDB ID: 6VK6) [42]. The secondary single oxygenation pathway was carried out in the presence of the cytochrome P450 enzyme of *Streptomyces avermitilis* (PDB ID: 3E5K) [37], alcohol dehydrogenase of *Escherichia coli* (PDB ID: 7BU3) [43], alcohol dehydrogenase of *Saccharomyces cerevisiae* (PDB ID: 1Q1N) [44], aldehyde dehydrogenase of *Staphylococcus aureus* (PDB ID: 6K10) [45], and aldehyde dehydrogenase of *Pseudomonas aeruginosa* (PDB ID: 4CAZ) [46]. The secondary hydroxylation pathway was conducted in the presence of the methane monooxygenase hydroxylase of *Methylococcus capsulatus* (PDB ID: 1FZI) [38], methane monooxygenase hydroxylase of *Methylosinus sporium* (PDB ID: 6D7K) [41], and methane monooxygenase hydroxylase of *Methylosinus trichosporium* OB3b (PDB ID: 6VK6) [42].

The degradation pathways of benzene and the types of degradation enzymes used were different from those used for the degradation of the other three petroleum hydrocarbon pollutants. The degraded proteins selected in each step of the benzene degradation path were identified: the hydroxylation pathway involved the participation of the methane monooxygenase hydroxylase of *Methylococcus capsulatus* (PDB ID: 1FZI) [38], methane monooxygenase hydroxylase of *Methylosinus trichosporium* (PDB ID: 1MHZ) [39], long-chain alkane monooxygenase of *Geobacillus thermodenitrificans* (PDB ID: 3B9N) [40], methane monooxygenase hydroxylase of *Methylosinus sporium* (PDB ID: 6D7K) [41], and methane monooxygenase hydroxylase of *Methylosinus trichosporium* OB3b (PDB ID: 6VK6) [42]. The secondary single oxygenation pathway involved the participation of five kinds of dioxygenases: procatechuic acid 2,3-dioxygenase of *Brevibacterium fuscum* (PDB ID: 2IGA) [47], catechol 1,2-dioxygenase of *Acinetobacter baylyi* ADP1 (PDB ID: 1DLT) [48], catechin acid 1,2-dioxygenase of *Pseudomonas putida* (PDB ID: 2AZQ) [49], 2,3-dioxygenase of *Pseudomonas alkylphenolica* (PDB ID: 3HPY) [50], and catechol 1,2-dioxygenase of *Acinetobacter radioresistens* OB3b (PDB ID: 2XSR) [51]. The hydrolysis pathway involved the participation of three kinds of hydrolases: epoxide hydrolase of *Aspergillus niger* (PDB ID: 3G02) [52], epoxide hydrolase of *Rhodococcus erythropolis* (PDB ID: 4XBT) [53], and epoxide hydrolase of *Bacillus megaterium* (PDB ID: 4IO0) [54]. The CoA pathway selected the benzene CoA reductase of *Geobacter metallireducens* GS-15 (PDB ID: 4Z3W) [55].

The proteins used for the studies were all from the PDB database (http://www.rcsb.org/pdb accessed on 9 June 2021).

### 2.2. Ability to Characterize Microbial Remediation of Petroleum Hydrocarbon Contaminated Sites Using the Molecular Dynamics Method

Previous studies have shown that the binding energy between receptors and molecules can indicate the degree of binding between them. The greater the absolute value of binding energy, the stronger the binding ability of the petroleum hydrocarbon degradation enzymes. In this paper, the Libdock docking module in Discovery Studio 2020 software version (Biovia, Waltham, MA, USA) was used to dock petroleum hydrocarbon contaminant molecules and study their degradation process intermediates. Protein molecules were pretreated prior to docking to remove ligands and water from the protein samples. According to the degradation path of each step and the degradation intermediates produced, the degradation intermediates were docked with the petroleum hydrocarbon degradation enzyme corresponding to each step of the degradation path. The docked complex of petroleum hydrocarbon degradation enzyme-petroleum hydrocarbon contaminant molecule/petroleum hydrocarbon contaminant degradation intermediate was moved to GROMACS 5.1.4 software version (University of Groningen, city, Groningen, the Netherlands) on the Dell PowerEdge R7425 server to conduct the molecular dynamics simulation calculations. The complex was placed in a periodic dodeca-cube with 10 nm-long sides in molecularly constrained spaces using a GROMOS96 43a1 force field. The Na^+^ ions were added to neutralize the charge of the system. The petroleum hydrocarbon degradation enzyme-petroleum hydrocarbon contaminant molecule/petroleum hydrocarbon contaminant degradation intermediate product molecule complex system was set up as a group and subjected to energy minimization simulations based on the steepest gradient method. The number of simulation steps was set at 50,000. Both the heat and pressure bath simulation times for the system were 100 ps, and the pressure bath size was set to a constant standard atmospheric pressure of 1 bar. The kinetic simulation operation time of different groups was set to 20 ps, and the binding energy of petroleum hydrocarbon degradation enzyme-petroleum hydrocarbon pollutant molecule/petroleum hydrocarbon pollutant degradation intermediate molecular complex system under different external environment stimulation was calculated.

### 2.3. Design of Inoculation Program for Petroleum Hydrocarbon Degradation Bacteria: Use of Java

For *n*-alkanes, branched-chain alkanes, and cycloalkanes, it is necessary to select a degradable protein that exhibits good degradability for the three petroleum hydrocarbon pollutants in each degradation path. Therefore, this requires the selection (via permutation and combination) of a variety of degraded proteins that can participate in each of the selected degradation paths to design the best inoculation plan for the petroleum hydrocarbon degradation bacteria. The single oxygenation pathway, hydroxylation pathway and secondary single oxygenation pathway involved the selection of one enzyme from each of the five enzymes. One enzyme was selected from the three enzymes in secondary hydroxylation pathway. A permutation and combination program was developed based on the Java language to simplify the calculation method (see the Appendix A for the specific program code).

For the design of the inoculation program of aromatic hydrocarbon degradation bacteria, the petroleum hydrocarbon degradation enzyme with the strongest binding ability was directly selected for inoculation and matching.

### 2.4. Optimization of the Microbial Remediation Inoculation Screening Method in Petroleum Hydrocarbon Contaminated Sites—Copeland Method

Copeland, A.H., University of Michigan, first proposed the Copeland Method in the seminar “On the Application of Mathematics in Social Sciences” [56]. It was initially applied to the field of election. Now it is widely used in logistics and transportation, event statistics, agriculture, and environmental science, among other fields. Caillaux et al. used the Copeland method to select the most suitable shipping route for container cargo between different coupling ports to save transportation cost and time [57]. Li et al. used the Copeland method to evaluate the ecological hazards of 61 environmental priority pollutants from two aspects of environmental behavior and biological toxicity index and predicted the ecological hazards of the new chemicals [58].

The core idea of the Copeland method is that “the minority is subordinate to the majority”. The calculation process is simple and convenient. Hence, this method was selected to design the best inoculation program for the petroleum hydrocarbon degradation bacteria. Firstly, each degradation enzyme in each degradation pathway was taken as the evaluation factor, and the degradation ability for each kind of petroleum hydrocarbon pollutant was taken as the evaluation index to determine the individual score. Following this, the evaluation scores of each degradation enzyme for the three kinds of petroleum hydrocarbon pollutants were added to obtain the overall score of each degradation enzyme. Finally, the overall evaluation score of each program was obtained by synthesizing the overall score of each degradation enzyme in the four-step degradation pathway. The best inoculation program for the petroleum hydrocarbon degradation bacteria was selected based on the score results. The specific evaluation steps of the Copeland method were determined: X evaluation factors were set, and each evaluation factor had N multiple evaluation indexes. The same evaluation index of each evaluation factor was compared in pairs. In the comparison process, the one with the good evaluation index was marked as +1, the one with the bad evaluation index was marked as −1, and the one with the same evaluation index was marked as 0. Following the evaluation of each of the evaluation factors and evaluation indices, the evaluation scores were added to find the final score. Its definition is as follows:$$T=\left\{\begin{array}{ccc} 1 & ; & N_{X_{a}}>N_{X_{b}}\\ 0 & ; & N_{X_{a}}=N_{X_{b}}\\ -1 & ; & N_{X_{a}}<N_{X_{b}} \end{array}\right.$$

Finally, the score T for each factor was added to determine the final ranking. When the sum of the scores of multiple factors is the same, the factor with the smallest variance is selected as the best program.

### 2.5. Verification of the Effect of the Best Inoculation Program for Petroleum Hydrocarbon Degradation Bacteria—The Molecular Dynamics Method

The degradation effect of the optimal inoculation program of petroleum hydrocarbon degradation bacteria was verified using the molecular docking method. Firstly, the four enzymes used in each program were docked by the ZDOCK module in Discovery Studio 2020 software version (Biovia, Waltham, MA, USA). The comprehensive petroleum hydrocarbon degradation enzymes were obtained by setting the RMSD Cut-off to cluster radius and Interface Cutoff to 9.0. ZRank was set to False to dock the four degradation enzymes one by one. The Cdocker module was then used to dock *n*-tetradecane, norphytane, cyclopentane, and their respective three intermediate degradation products to obtain the Cdocker energy values to characterize the overall degradation capacity of the four degradation enzymes.

## 3. Results and Discussion

### 3.1. Identification of the Main Degradation Pathways for Microbial Degradation of Petroleum Hydrocarbon Pollutants

It has been previously reported that the microbial degradation of straight-chain alkanes is primarily carried out via four metabolic pathways: single-terminal oxidation, double-terminal oxidation, sub-terminal oxidation, and direct dehydrogenation [59]. Branched-chain alkanes are primarily degraded through the single-terminal oxidation and double-terminal oxidation processes. Some microorganisms degrade branched-chain alkanes through the sub-terminal oxidation pathway [60]. Cycloalkanes are mainly metabolized through the processes of single-terminal oxidation, sub-terminal oxidation, and direct dehydrogenation [61]. When microorganisms degrade aromatic hydrocarbons, they first degrade the aromatic hydrocarbons into catechins, and then break the benzene ring through the processes of orthotopic cracking and eccentricity cracking under the action of dioxygenase [62]. In this paper, the representative contaminant molecules, *n*-tetradecane, norphytane, cyclopentane, and benzene, were selected as the research objects. The molecular structures of the pollutants were downloaded from the PubChem molecular library (http://pubchem.ncbi.nlm.nih.gov accessed on 7 June 2021). As the degradation pathway of the petroleum hydrocarbon pollutants is complex, we studied the main steps in the degradation pathway. The main degradation pathways for *n*-tetradecane, norphytane, cyclopentane, and benzene are shown in Figure 1, Figure 2, Figure 3 and Figure 4.

Firstly, under the action of monooxygenase, *n*-tetradecane is single-terminally oxidized, and the terminal methyl group is oxidized to hydroxyl to generate tridecane-1-ol. Following this, under the action of hydroxylase, tridecane-1-ol is oxidized to dodecan-1-ol, and acetic acid is generated. Under the action of monooxygenase/alcohol dehydrogenase and aldehyde dehydrogenase, dodecan-1-ol is oxidized to hexane-1-ol, and three molecules of acetic acid is generated. Finally, hexan-1-ol is oxidized to adipic acid under the action of hydroxylase.

For norphytane, single-terminal oxidation is performed on norphytane under the action of monooxygenase, and the terminal methyl group is oxidized to hydroxyl to generate 2,6,10,14-tetramethylpentadecane-1-alcohol. Following this, under the action of hydroxylase, 2,6,10,14-tetramethylpentadecan-1-ol is oxidized to 2,6,10-trimethylpentadecane-1,14-diol, and under the repeated action of monooxygenase/alcohol dehydrogenase and aldehyde dehydrogenase, 2,6,10-trimethylpentadecane-1,14-diol is oxidized to 2-methyl-1-propanol. Finally, under the action of hydroxylase, 2-methyl-1-propanol is oxidized to 2-methyl-1,3-propanediol.

Cyclopentane is firstly oxidized at one end under the action of monooxygenase. The terminal methyl group is oxidized to the hydroxyl group to generate cyclopentanol. Under the action of hydroxylase, cyclopentanol is oxidized to cyclopentanone, and under the action of monooxygenase/alcohol dehydrogenase and aldehyde dehydrogenase, cyclopentanone is oxidized to dihydrofuran-2(3H)-one. Finally, under the action of hydroxylase, dihydrofuran-2(3H)-one is oxidized to 3-hydroxypropionic acid.

Benzene is first degraded to catechol under the action of hydroxylase. Following this, under the action of catechol 1,2/2,3 dioxygenase, the benzene ring in catechol is broken to form (Z)-hexa-3-enedione acid. The produced (Z)-hexa-3-enedione acid is converted to 3-oxoadipate under the action of hydrolase. Finally, 3-oxoadipate is converted to succinic acid by the action of benzene-CoA transferase.

As petroleum hydrocarbon degradation is a very complicated process, it usually takes dozens or even hundreds of steps to degrade petroleum hydrocarbon pollutants into non-toxic and harmless water and carbon dioxide molecules. The bioconcentration and long-distance migration of the final product were predicted using the EPI database. The results are shown in Table 1. The bioconcentration factor log*BCF* [67] was used to characterize petroleum hydrocarbon pollutants and the bioconcentration of the degradation products. The larger the log*BCF* value, the substance is more likely to accumulate in the organism. In other words, the substance is more biohazardous. The N-octanol-air partition coefficient (log*K*_OA_) [68] was used to characterize the long-distance migration of petroleum hydrocarbon pollutants and their degradation products. The larger the log Koa value, the easier it is for the substance to migrate over long distances. Under these conditions, it is difficult to carry out centralized degradation and treatment. We observed that, compared with the original petroleum hydrocarbon pollutants, the bioconcentration and long-distance migration of the final product were significantly reduced. Therefore, only the single oxygenation pathway, hydroxylation pathway, secondary single oxygenation pathway, and secondary hydroxylation pathway were selected as the main degradation pathways of n-alkanes, branched-chain alkanes, and cycloalkanes. The hydroxylation pathway, 9eoxygenation pathway, hydrolysis pathway, and CoA pathway were selected as the main degradation pathways of aromatic hydrocarbons. The subsequent degradation pathways were not considered for the studies.

### 3.2. Calculation of Binding Energy of the Degraded Protein of Petroleum Hydrocarbon Degradation Bacteria and Petroleum Hydrocarbon Pollutants

We used the GROMACS 5.1.4 software version (University of Groningen, city, Groningen, the Netherlands) to perform the molecular dynamics simulation calculations on the Dell PowerEdge R7425 server. The binding energy of the degraded protein and petroleum hydrocarbon pollutants under each degradation pathway were calculated to characterize the degradation ability of degradation enzymes. A study of the main degradation pathways identified by us revealed that each degradation pathway corresponded to different degradation enzymes and intermediate degradation products. The calculation results of the binding energy of the degraded protein and petroleum hydrocarbon pollutants and the corresponding intermediate degradation products under each degradation path are shown in Table 2, Table 3, Table 4, Table 5 and Table 6.

The binding energy of each degraded protein with petroleum hydrocarbon pollutants can be used to understand the degradation ability of the degraded protein. As shown in Table 2, under conditions of the single oxygenation pathway, the cytochrome P450 enzyme 3E5K exhibits the strongest degradation ability for *n*-tetradecane and norphytane. The bacterial cytochrome P450 enzyme 1SMJ exhibits the strongest degradation ability for cyclopentane. As shown in Table 3, under conditions of the hydroxylation pathway, the methane monooxygenase hydroxylase 1FZI exhibits the strongest degradation ability for tridecane-1-ol, the methane monooxygenase hydroxylase 1MHZ exhibits the strongest degradation ability for the degradation products of 2,6,10,14-tetramethylpentadecan-1-ol, and the methane monooxygenase hydroxylase 6D7K exhibits the strongest degradation ability for cyclopentanol. As shown in Table 4, under conditions of the secondary single oxygenation pathway, the aldehyde dehydrogenase 4CAZ exhibits the strongest degradation ability for dodecan-1-ol and cyclopentanone and cytochrome P450 enzyme 3E5K exhibits the strongest degradation ability for 2,6,10-trimethylpentadecane-1,14-diol. As shown in Table 5, under the conditions of the secondary hydroxylation pathway, the methane monooxygenase hydroxylase 6D7K exhibits the strongest degradation ability for hexan-1-ol, the methane monooxygenase hydroxylase 6VK6 exhibits the strongest degradation ability for 2-methyl-1-propanol, and the methane monooxygenase hydroxylase 1FZI exhibits the strongest degradation ability for dihydrofuran-2(3H)-one. As shown in Table 6, the strongest ability to degrade benzene is exhibited by the methane monooxygenase hydroxylase 6VK6 under the conditions of the hydroxylation pathway. The strongest ability to degrade catechol is exhibited by the catechol 1,2-dioxygenase 1DLT under the conditions of the double oxygenation pathway. The strongest ability to degrade (Z)-hexa-3-enedione acid is exhibited by the epoxide hydrolase 4XBT under the conditions of the hydrolysis pathway. Under conditions of the CoA pathway, only the benzene CoA reductase 4Z3W was selected to degrade 3-oxohexanedioic acid.

Figure 5a–d show the heat maps of the degradation ability of various degradation enzymes. Based on the above results for each petroleum hydrocarbon pollutant, it can be concluded that the enzymes with the best degradation efficiency vary from pathway to pathway. Therefore, the comprehensive degradation ability of the degradation enzymes corresponding to each degradation path should be comprehensively studied to obtain an inoculation program that exhibits high degradability for all three petroleum hydrocarbon pollutants.

### 3.3. Design of Inoculation Program for Petroleum Hydrocarbon Degradation Bacteria Based on the Java Language: A Permutation and Combination Program

For the four-step degradation pathway, one of the five monooxygenases in the monooxygenase pathway, one of the five hydroxylases in the hydroxylation pathway, one of the five monooxygenases/aldehyde dehydrogenases/alcohol dehydrogenases in the secondary monooxygenation pathway, and one of the enzymes from the three hydroxylases in the secondary hydroxylation pathway for permutation and combination were selected to formulate the inoculation plan for the petroleum hydrocarbon degradation bacteria that degrade normal alkanes, branched-chain alkanes, and cycloalkanes. The designed program was used for the permutation and combination studies of the inoculation programs of petroleum hydrocarbon degradation bacteria. Finally, 375 inoculation programs of petroleum hydrocarbon degradation bacteria were obtained. The number of each petroleum hydrocarbon degradation enzyme is shown in Table 6. Although the hydroxylation pathway and the secondary hydroxylation pathway involve the presence of the same degradation enzymes, the degradation products of petroleum hydrocarbons degraded following the two degradation pathways are different. The two parts of the same degradation enzymes were numbered in two ways to explore the degradation ability of these enzymes.

### 3.4. Determination of the Optimal Inoculation Program of Petroleum Hydrocarbon Degradation Bacteria Based on the Copeland Method

In this paper, the Copeland method was used to determine the best petroleum hydrocarbon degradation bacteria inoculation plan for the degradation of *n*-alkanes, branched-chain alkanes, and cycloalkanes. First, the three petroleum hydrocarbon pollutants were evaluated one by one in the presence of each degradation enzyme for each degradation path. Following this, the evaluation scores of the three petroleum hydrocarbon pollutants were summed to obtain the comprehensive degradability evaluation score for each enzyme. The comprehensive degradation ability evaluation score of each degradation enzyme was substituted into 375 petroleum hydrocarbon degradation bacteria inoculation programs for program evaluation, and the evaluation score of each program was obtained to select the best petroleum hydrocarbon degradation bacteria inoculation program. The evaluation scores of 375 kinds of petroleum hydrocarbon degradation bacteria inoculation programs are shown in Figure 2.

As shown in Figure 6, programs [4, 7, 15, 16] and [4, 10, 15, 16] exhibit the highest evaluation scores, both of which are 22. No. 4, No. 15, and No. 16 degradation enzymes were selected for the other three steps in the two groups. To screen out the best petroleum hydrocarbon degradation bacteria inoculation plan, the binding energies of No. 7 and No. 10 degradation enzymes and petroleum hydrocarbon pollutants in the hydroxylation path in these two groups of plans were calculated. The best inoculation program was determined by comparing the variance of degradation enzymes in the hydroxylating pathway. The sample variance calculation formula is represented as follows:$$S^{2}=\frac{\sum {\left(X-\bar{X}\right)}^{2}}{n-1}$$

The variances of degradation enzymes No. 7 and No. 10 of the hydroxylation pathway with petroleum hydrocarbon pollutants are shown in Table 7. It can be seen from Table 7 that the variance of the program [4, 10, 15, 16] is the least. Therefore, program [4, 10, 15, 16] is the best program for the inoculation of petroleum hydrocarbon-degradation bacteria.

Therefore, the optimal combination of the degradation enzymes for *n*-alkanes, branched-chain alkanes, and cycloalkanes was as follows: the single oxygen route involved the participation of the cytochrome P450 enzyme (PDB ID: 3E5K) from *Streptomyces avermitilis*, the hydroxylation path used the methane monooxygenase hydroxylase of *Methylosinus trichosporium* OB3b (PDB ID: 6VK6), and the second single oxygenation route used the aldehyde dehydrogenase (PDB ID: 4CAZ) of *Pseudomonas aeruginosa*. The methane monooxygenase hydroxylase (PDB ID: 1FZI) of *Methylococcus capsulatus* was selected for secondary hydroxylation. The combination of the degradation enzyme to realize the best petroleum hydrocarbon degradation bacteria inoculation plan for the degradation of aromatic hydrocarbons is as follows: Methane monooxygenase hydroxylase (PDB ID: 6VK6) of *Methylosinus trichosporium* OB3B was selected for the hydroxylation pathway, catechol 1, 2-dioxygenase (PDB ID: 6VK6) of *Acinetobacter baylyi* ADP1 was selected for the 14eoxygenation pathway (PDB ID: 1DLT), epoxide hydrolase of *Rhodococcus erythropolis* (PDB ID: 4XBT) was used for hydrolysis, and benzene CoA reductase of *Geobacter metallireducens* GS-15 (PDB ID: 4Z3W) was used for the CoA route.

In summary, the optimal inoculation plan for petroleum hydrocarbon degradation bacteria was determined. The single oxygenation path primarily involved the participation of *Streptomyces avermitilis*, the hydroxylation path primarily involved the participation of *Methylosinus trichosporium* OB3b, and the secondary single oxygenation path primarily involved the participation of *Pseudomonas aeruginosa*. The secondary hydroxylation path involves the participation of *Methylococcus capsulatus*, the double oxygenation path involves the activity of *Acinetobacter baylyi* ADP1, the hydrolysis path primarily involves the participation of *Rhodococcus erythropolis*, and the CoA path primarily involves the participation of *Geobacter metallireducens* GS-15.

### 3.5. Verification of the Degradation Effect of the Optimal Inoculation Program of Petroleum Hydrocarbon Degradation Bacteria Based on the Molecular Docking Method

The degradation effect of the petroleum hydrocarbon degradation bacteria inoculation program was verified using the Cdocker module in Discovery Studio 2020 software version (Biovia, Waltham, MA, USA). The inoculation program of petroleum hydrocarbon degradation bacteria in the first 13 points of the Copeland method was selected to simplify the calculation process and verify the degradation effect. The Cdocker energy values of the 13 inoculation programs of petroleum hydrocarbon degradation bacteria are shown in Figure 3.

As can be seen from Figure 7, the absolute Cdocker energy value of the best inoculation program [4, 10, 15, 16] is the maximum, indicating that the comprehensive degradation effect of this program is the best. The linear fit between the degradation effect of the remaining 12 petroleum hydrocarbon degradation bacteria and their Copeland method score rank is good (Pearson’s r = 0.907 > 0.9). This indicates that the scores of the petroleum hydrocarbon degradation bacteria inoculated by the Copeland method are positively proportional to the degradation effect of the petroleum hydrocarbon pollutants, confirming the best inoculation program of the petroleum hydrocarbon degradation bacteria. This confirms the rationality of the optimal inoculation program determined using the Copeland method.

### 3.6. Prospect of the Optimal Inoculation Program of Petroleum Hydrocarbon Degradation Bacteria

The optimal petroleum hydrocarbon degradation bacteria inoculation program devised in this paper yielded good results at the theoretical simulation level. *Streptomyces avermitilis*, *Methylosinus trichosporium* OB3b, *Pseudomonas aeruginosa, Methylococcus capsulatus, Acinetobacter baylyi* ADP1, *Rhodococcus erythropolis*, and *Geobacter metallireducens* GS-15 of the optimal petroleum hydrocarbon degradation bacteria inoculation program have all been used individually in the remediation of actual petroleum contaminated sites and have been effective in the remediation of petroleum hydrocarbon contaminated sites [30]. The best petroleum hydrocarbon degradation bacteria inoculation program designed in this paper will put into practical remediation application, and the combined application of these degradation bacteria to remediate petroleum contaminated sites is the next step of our research. The theoretical scheme designed in this paper will put into practical application to achieve practical results. In practical applications, we will further refine the inoculation conditions to minimize the impact of uncertainties. In the practical application process, the inoculated petroleum hydrocarbon degradation bacteria will rapidly degrade petroleum hydrocarbon pollutants, and the content of petroleum hydrocarbon pollutants in soil will decrease rapidly due to the degradation of petroleum hydrocarbon degradation bacteria [69], and the uniformity and abundance of petroleum hydrocarbon degradation bacteria will tend to be stable as time goes by [70]. The degradation function of petroleum hydrocarbon degradation bacteria can continue to exist for a long time when the external conditions are controlled and the bacteria are in a suitable survival condition. When the uniformity and abundance of petroleum hydrocarbon degradation bacteria decreases, it could be adjusted by adjusting the external conditions or adding the nutrients necessary for petroleum hydrocarbon degradation bacteria to restore the good survival state; if necessary, it could also be selected to re-inoculate a small number of bacteria to adjust the proportion of bacteria in the field.

## 4. Conclusions

The degradation path of the petroleum hydrocarbon pollutants by petroleum hydrocarbon degradation bacteria was analyzed. A variety of degradation enzymes corresponding to each step were selected for the main degradation path, and the binding energy of each degradation enzyme was calculated suing the molecular dynamics method to characterize the degradation ability. We used the developed Java program to identify the degradation enzymes in each step of the degradation path to obtain multiple groups of petroleum hydrocarbon degradation bacteria inoculation programs. The permutation and combination method was used for the same. The Copeland method was used to determine the best petroleum hydrocarbon degradation bacteria inoculation program. The best petroleum hydrocarbon degradation bacteria inoculation program can effectively repair the petroleum hydrocarbon pollution in the site and provide a reliable theoretical program for the bioremediation of petroleum hydrocarbon contaminated sites.

## Figures and Tables

**Figure 1 ijerph-18-08794-f001:**
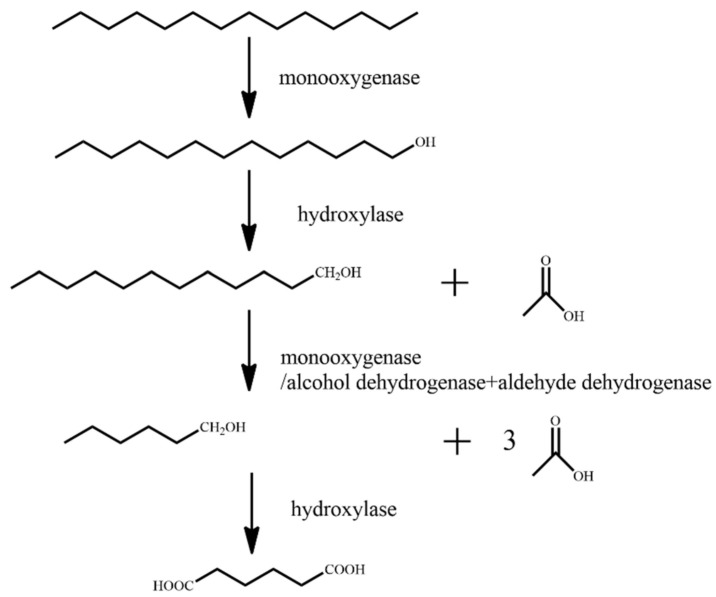
Degradation pathway of *n*-tetradecane [63].

**Figure 2 ijerph-18-08794-f002:**
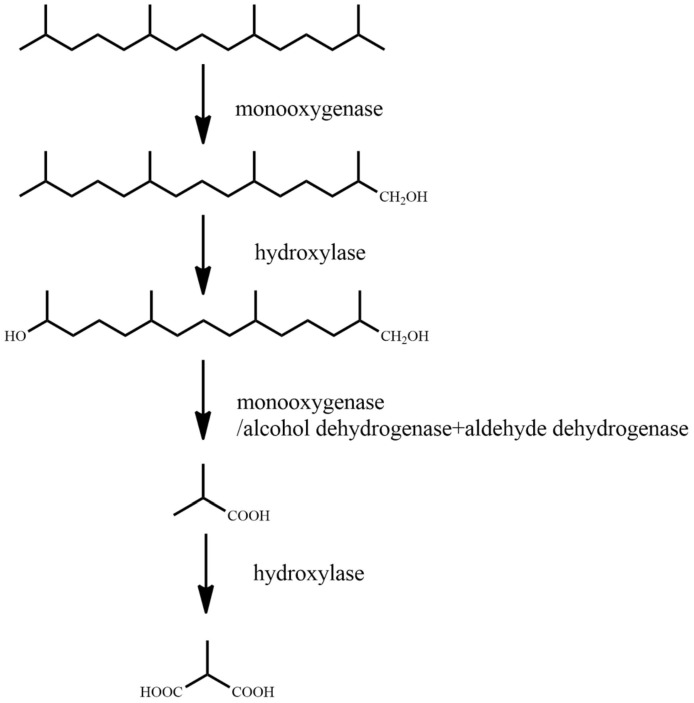
Degradation pathway of norphytane [64].

**Figure 3 ijerph-18-08794-f003:**
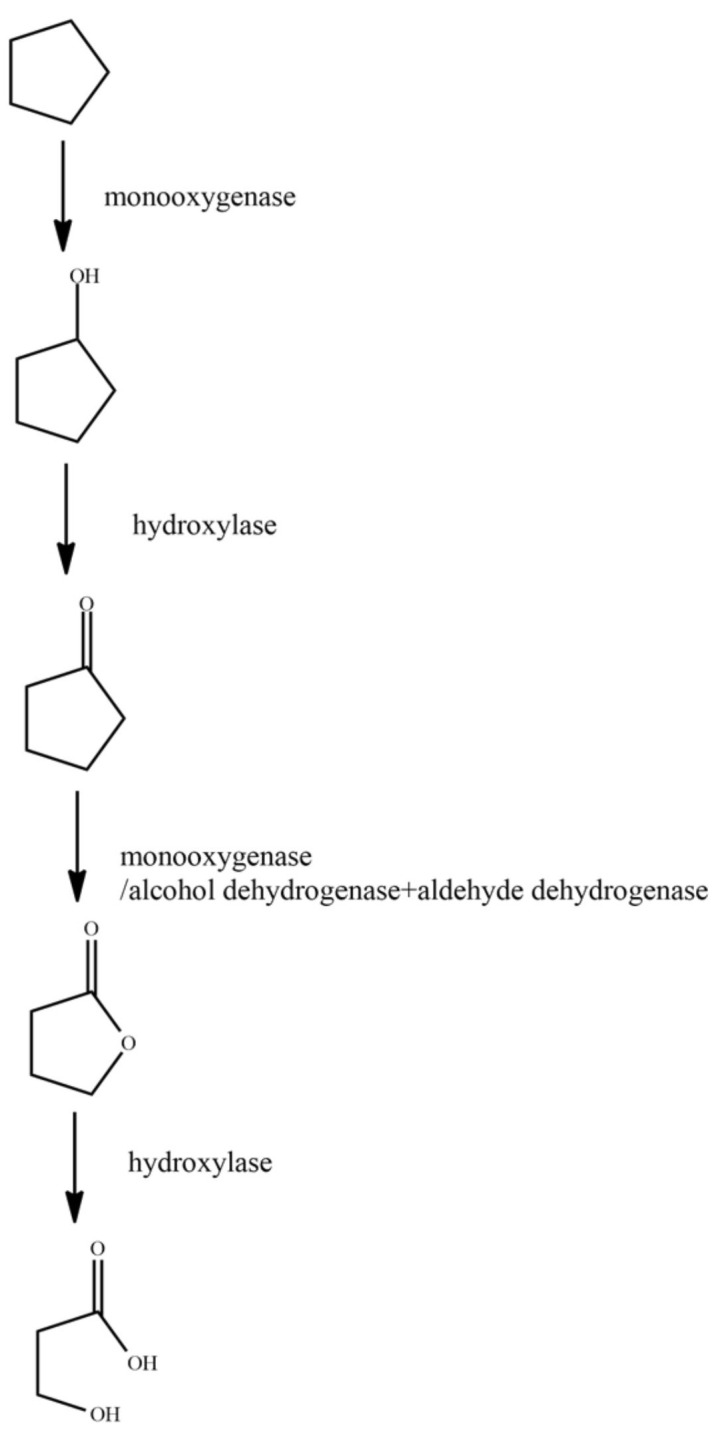
Degradation pathway of cyclopentane [65].

**Figure 4 ijerph-18-08794-f004:**
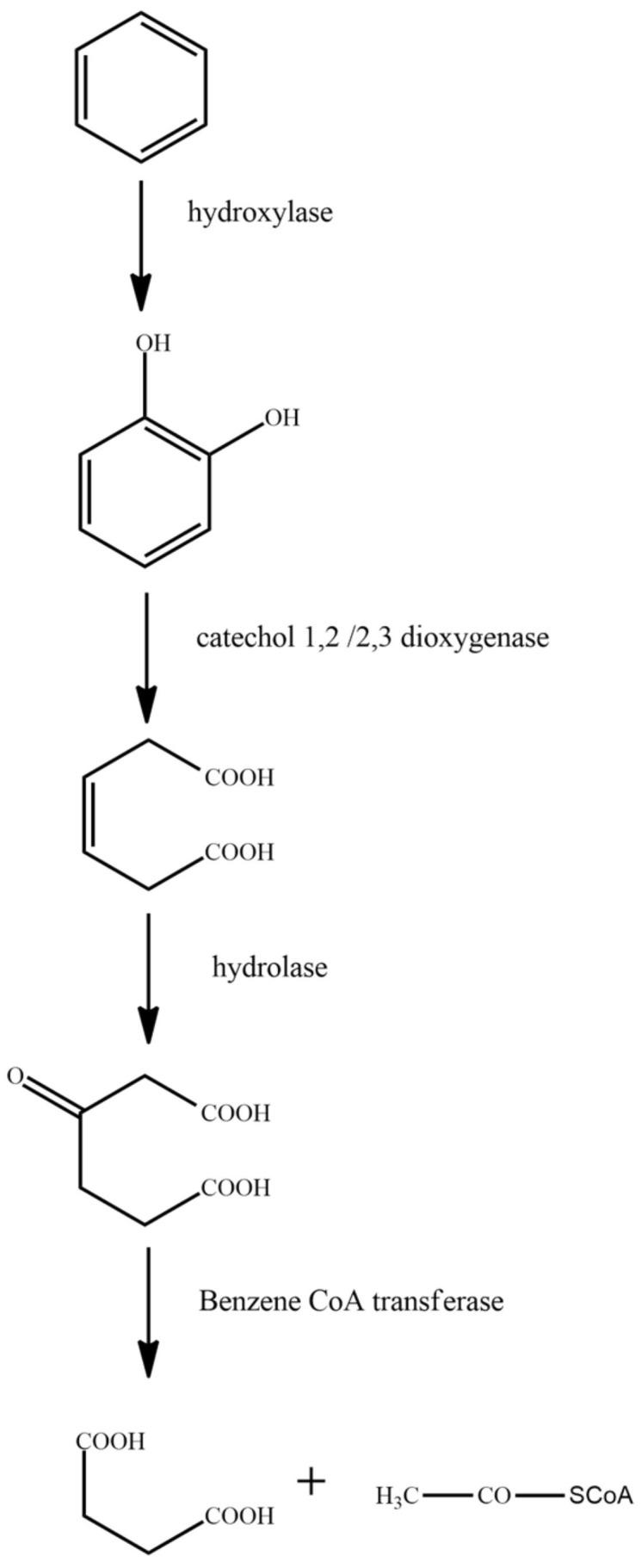
Degradation pathway of benzene [66].

**Figure 5 ijerph-18-08794-f005:**
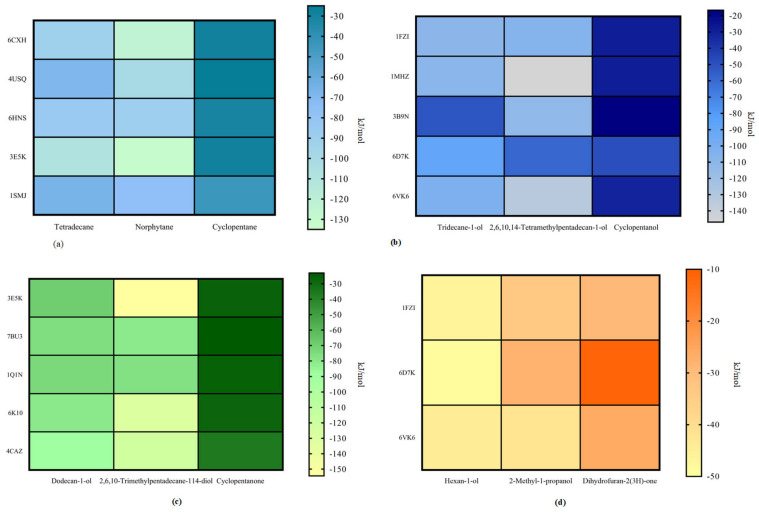
Degradability of petroleum hydrocarbon pollutants and intermediate degradation products by each degradation enzymes (**a**) single oxygenation pathway, (**b**) hydroxylation pathway, (**c**) single oxygenation pathway, and (**d**) secondary hydroxylation.

**Figure 6 ijerph-18-08794-f006:**
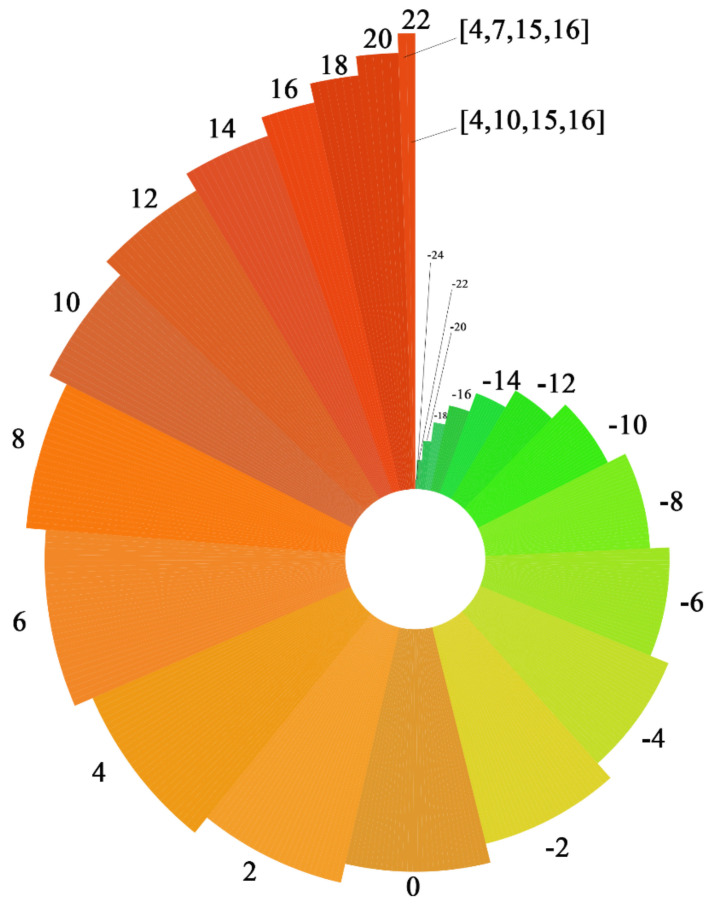
Score for 375 petroleum hydrocarbon degradation bacteria inoculation programs.

**Figure 7 ijerph-18-08794-f007:**
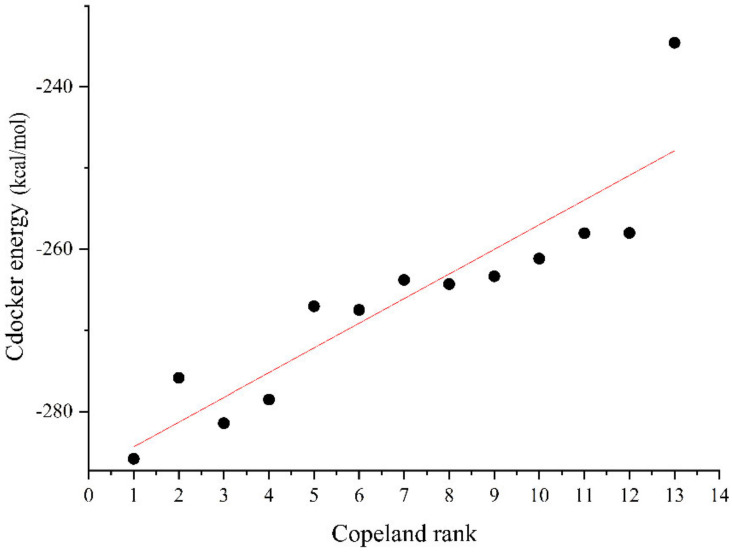
Plot of the score rankings of the 13 petroleum hydrocarbon degradation bacteria inoculation programs against the Cdocker energy values.

**Table 1 ijerph-18-08794-t001:** Comparison of bioconcentration and long-distance migration of petroleum hydrocarbon pollutants and their degradation products.

Petroleum Hydrocarbon Pollutants	log*BCF* [63]	log*K*_OA_ [64]	Degradation Products	log*BCF* [63]	log*K*_OA_ [64]
*n*-Tetradecane	3.430	4.625	Adipic acid	0.500	9.795
Norphytane	2.362	5.937	2-Methyl-1, 3-propanediol	0.500	8.825
Cyclopentane	1.646	2.207	3-Hydroxypropionic acid	0.500	8.074
Benzene	1.072	2.780	Succinic acid	0.500	10.246

**Table 2 ijerph-18-08794-t002:** Calculation results of the binding energy of each degraded protein and petroleum hydrocarbon pollutants under single oxygenation pathway.

Degradation Enzyme	*n*-Tetradecane	Norphytane	Cyclopentane
6CXH	−91.013	−124.464	−28.367
4USQ	−69.372	−100.321	−25.052
6HNS	−86.015	−90.152	−30.944
3E5K	−108.902	−131.824	−28.575
1SMJ	−66.647	−75.650	−44.472

**Table 3 ijerph-18-08794-t003:** Calculation results of binding energy of each degraded protein and intermediate degradation products of petroleum hydrocarbon pollutants under hydroxylation pathway.

Degradation Enzyme	Tridecane-1-ol	2,6,10,14- Tetramethylpentadecan-1-ol	Cyclopentanol
1FZI	−110.404	−106.555	−28.650
1MHZ	−109.605	−146.983	−28.647
3B9N	−51.817	−112.476	−16.685
6D7K	−88.037	−58.433	−49.200
6VK6	−103.004	−132.869	−31.266

**Table 4 ijerph-18-08794-t004:** Calculation results of binding energy of each degraded protein and intermediate degradation products of petroleum hydrocarbon pollutants under secondary single oxygenation pathway.

Degradation Enzyme	Dodecan-1-ol	2,6,10-Trimethylpentadecane-1,14-diol	Cyclopentanone
3E5K	−69.416	−154.453	−26.549
7BU3	−75.925	−80.793	−23.091
1Q1N	−73.881	−76.893	−26.337
6K10	−80.342	−130.382	−27.720
4CAZ	−88.832	−121.662	−35.881

**Table 5 ijerph-18-08794-t005:** Calculation results of binding energy of each degraded protein and intermediate degradation products of petroleum hydrocarbon pollutants under secondary hydroxylation pathway.

Degradation Enzyme	Hexan-1-ol	2-Methyl-1-Propanol	Dihydrofuran-2(3H)-one
1FZI	−46.861	−33.517	−29.305
6D7K	−49.145	−27.987	−11.258
6VK6	−44.197	−41.865	−26.201

**Table 6 ijerph-18-08794-t006:** Number of petroleum hydrocarbon degradation enzymes.

NO.	Enzyme	NO.	Enzyme	NO.	Enzyme	NO.
1	6CXH	7	1MHZ	13	1Q1N	1
2	4USQ	8	3B9N	14	6K10	2
3	6HNS	9	6D7K	15	4CAZ	3
4	3E5K	10	6VK6	16	1FZI	4
5	1SMJ	11	3E5K	17	6D7K	5

**Table 7 ijerph-18-08794-t007:** Variance of degradation enzyme in the second step degradation pathway.

No.	7	10
Variance	2439.414	1817.936

## Data Availability

The data presented in this study are available contained within the article.

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
