# Peer review of "Design of a Microbial Remediation Inoculation Program for Petroleum Hydrocarbon Contaminated Sites Based on Degradation Pathways"

_ijerph, 2021, doi:10.3390/ijerph18168794_

Round 1

Reviewer 1 Report

1) Show important numerical values in the abstract

2) What are the different biochemical reactions involved in the degradation process of petroleum contaminated sites. Give all the reactions, pathways and end products + enzymes involved. Refer to literature and do a good review of this. Without this aspect, we CANNOT understand anything.

3) Compare the regulations concerning Petroleum hydrocarbon in your country, USA and also from other ASIAN countries.

4) Give examples and case studies from  petroleum hydrocarbon contaminated sites  in your country and give examples how it was remediated.

5) Provide details of all the control and blank test sites.

6) Did Cycloalkanes exhibit toxic effects?

7) What was the rate limiting steps in this process?

8) What are the optimum conditions for the action of monooxygenase/alcohol dehydrogenase and aldehyde dehydrogenase,?

9) Units are missing in the graphs.

10) Write the practical applications of this work in 200 words + write the future research prospects in 100 words, before the conclusions.

11) Check the reference formatting mistakes.

Reviewer 2 Report

This manuscript

 investigate microbial remediation inoculation for petroleum hydrocarbon contaminated sites based on degradation pathways. The  degradation pathways of petroleum hydrocarbon degradation bacteria were analyzed. In general, the manuscript is well designed, the materials and methods are well presented, the presentation of the results together with the discussion bring clear information to the readers. I only have a question regarding the application of that microbial inoculation in a real systems, how long this inoculation can stay in the real systems and still have a function? The authors need to discuss more on this aspect.
